# Parkinson’s Disease and SARS-CoV-2 Infection: Particularities of Molecular and Cellular Mechanisms Regarding Pathogenesis and Treatment

**DOI:** 10.3390/biomedicines10051000

**Published:** 2022-04-26

**Authors:** Aurelian Anghelescu, Gelu Onose, Cristina Popescu, Mihai Băilă, Simona Isabelle Stoica, Ruxandra Postoiu, Elena Brumă, Irina Raluca Petcu, Vlad Ciobanu, Constantin Munteanu

**Affiliations:** 1Faculty of Midwives and Nursing, University of Medicine and Pharmacy “Carol Davila”, 020022 Bucharest, Romania; aurelian.anghelescu@umfcd.ro (A.A.); stoica.simona@umfcd.ro (S.I.S.); 2Neuromuscular Rehabilitation Clinic Division, Teaching Emergency Hospital “Bagdasar-Arseni”, 041915 Bucharest, Romania; cristina_popescu_recuperare@yahoo.com (C.P.); baila_mihai@yahoo.com (M.B.); postoiu.ruxandra@yahoo.ro (R.P.); brumaelena@ymail.com (E.B.); 3Faculty of Medicine, University of Medicine and Pharmacy “Carol Davila”, 020022 Bucharest, Romania; 4Physical Medicine and Rehabilitation Laboratory (Treatment Base), Turnu Magurele Municipal Hospital, 145200 Turnu Magurele, Romania; varsa_raluca@yahoo.com; 5Computer Science Department, Politehnica University of Bucharest, 060042 Bucharest, Romania; vladutc83@gmail.com; 6Department of Biomedical Sciences, Faculty of Medical Bioengineering, University of Medicine and Pharmacy “Grigore T. Popa” Iași, 700454 Iași, Romania

**Keywords:** COVID-19, SARS-CoV-2, neuroinflammation, neurodegenerative diseases, Parkinson’s disease, pathogenesis, genetics, epigenetic, nanotechnology, therapeutic management

## Abstract

Accumulating data suggest that chronic neuroinflammation-mediated neurodegeneration is a significant contributing factor for progressive neuronal and glial cell death in age-related neurodegenerative pathology. Furthermore, it could be encountered as long-term consequences in some viral infections, including post-COVID-19 Parkinsonism-related chronic sequelae. The current systematic review is focused on a recent question aroused during the pandemic’s successive waves: are there post-SARS-CoV-2 immune-mediated reactions responsible for promoting neurodegeneration? Does the host’s dysregulated immune counter-offensive contribute to the pathogenesis of neurodegenerative diseases, emerging as Parkinson’s disease, in a complex interrelation between genetic and epigenetic risk factors? A synthetic and systematic literature review was accomplished based on the ”Preferred Reporting Items for Systematic Principles Reviews and Meta-Analyses” (PRISMA) methodology, including registration on the specific online platform: International prospective register of systematic reviews—PROSPERO, no. 312183. Initially, 1894 articles were detected. After fulfilling the five steps of the selection methodology, 104 papers were selected for this synthetic review. Documentation was enhanced with a supplementary 47 bibliographic resources identified in the literature within a non-standardized search connected to the subject. As a final step of the PRISMA method, we have fulfilled a Population-Intervention-Comparison-Outcome-Time (PICOT)/Population-Intervention-Comparison-Outcome-Study type (PICOS)—based metanalysis of clinical trials identified as connected to our search, targeting the outcomes of rehabilitative kinesitherapeutic interventions compared to clinical approaches lacking such kind of treatment. Accordingly, we identified 10 clinical trials related to our article. The multi/interdisciplinary conventional therapy of Parkinson’s disease and non-conventional multitarget approach to an integrative treatment was briefly analyzed. This article synthesizes the current findings on the pathogenic interference between the dysregulated complex mechanisms involved in aging, neuroinflammation, and neurodegeneration, focusing on Parkinson’s disease and the acute and chronic repercussions of COVID-19. Time will tell whether COVID-19 neuroinflammatory events could trigger long-term neurodegenerative effects and contribute to the worsening and/or explosion of new cases of PD. The extent of the interrelated neuropathogenic phenomenon remains obscure, so further clinical observations and prospective longitudinal cohort studies are needed.

## 1. Introduction

The actual pandemic has significant global health consequences. Starting from recent concerns that COVID-19 might challenge new waves of neurodegenerative diseases (NDs) in susceptible patients, this review summarizes the current understanding of the complex pathogenic mechanisms and interferences at the cellular and molecular scales between the main “film actors” in the pandemic era: “the good, the bad and the ugly“, meaning the natural immune responses and homeostatic mechanisms, the complex pathways of neuroinflammation and neurodegeneration focussing Parkinson’s disease (PD), and SARS-CoV-2.

COVID-19 induces heterogeneous clinical manifestations and multisystemic organ damage and complications, including neurological, psychiatric, psychological, and psychosocial impairments. More than half of patients experienced at least one neurological symptom: headache, nausea, vomiting, generalized seizure and altered consciousness, strokes of multiple etiologies, and immune-mediated Guillain-Barre syndrome. The mortality rate of patients with associated neurological complications was double compared to those without neurological complications, and they needed more intensive care assistance [1]. During the successive waves of the pandemic, a contemporary question emerged: are post COVID-19 immune-mediated reactions the culprit for promoting neurodegeneration, or does the organism’s insufficient/aberrant immune riposte contribute to the pathogenesis of NDs, including emerging PD?

The acute neurological and neuropsychiatric repercussions of COVID-19, respectively, the variety of chronic post-infectious persistent symptoms, such as the “long-haul/post-COVID” syndrome [2,3,4,5,6,7,8,9,10,11], were the subject of thousands of studies all over the world [11,12,13,14]. Accumulating literature supports the critical role of a dysregulated immune system in promoting persistent neuroinflammation, creating the pathogenic trajectory of the NDs such as Alzheimer’s and Parkinson’s disease [4,6,15]. NDs are a group of heterogeneous disorders characterized by slow and progressive alteration of the neuronal cellular pool to their death, occurring in specific regions of the central nervous system (CNS), paralleled by a progressive decline of cerebral cognitive functions, leading to severe sensory-motor impairments [16].

Accurate animal models for Parkinson’s and Alzheimer’s diseases are not available. Studies investigating the complex cellular and molecular processes during the natural history of NDs use brain organoids, which are in vitro simplified model systems of organs generated from pluripotent stem cells [17].

Aging is a natural progressive biological involution process associated with molecular and cellular phenotypical hallmarks: genomic instability, reduced telomere length, epigenetic alterations, mitochondrial dysfunction, loss of proteostasis, cellular senescence, stem cell exhaustion, and altered intercellular communication [18]. Older age is the most common risk factor for developing NDs and one of the top risks for getting severe COVID-19 infection. In addition, chronic neuroinflammation-mediated neurodegeneration is considered a significant contributing factor to neuronal death and is associated with age-related brain disorders.

## 2. Methods

We followed the PRISMA methodology tackling all its standardized steps to achieve this synthetic and systematic literature review. Specifically, we have investigated, using specific keyword combinations/“syntaxes” (see Appendix A), five renowned international medical databases: Elsevier, National Center for Biotechnology Information (NCBI)/PubMed, NCBI/PubMed Central (PMC), Physiotherapy Evidence Database (PEDro), and ISI Web of Science, the last being used to determine whether a work selected through our PRISMA search was published in an ISI-indexed journal.

The period investigated was between 1 January 2021 and 31 December 2021. We have considered only written in English and free full-text available papers. Then, we registered this systematic literature review on the dedicated online platform International Prospective Register of Systematic Reviews—PROSPERO, no. 312183. Accordingly, after the above-mentioned first step—i.e., the primary collection of all the articles found in the international medical databases—we removed the duplicates in the second step. We verified and kept only the ISI-indexed publications in the third one.

After, in the fourth step, we made a preliminary indirect evaluation of the scientific impact/quality of the retained papers availing a PEDro classification/scoring inspired method, i.e., accepting as qualified only the works having at least four citations, thus corresponding to the PEDro score paradigm (“fear quality = PEDro score 4–5”).

In the fifth step, we have done the qualitative analysis of the articles retained at this stage of filtering, precisely 136 articles. Within this step, we have eliminated those papers that, although eligible through the criteria above, the content of their full text appeared irrelevant or non-contributive to the subject we had approached (“full-text articles excluded with reasons”). Therefore, after this minute selection endeavor, we obtained, through a final quantitative and qualitative selection, 104 papers from which we have extracted the essential bibliographic resources afferent to this article (Figure 1—Our completed PRISMA type of flow diagram; Appendix A presents the citation data of the eventually selected papers in our systematic literature review).

Additionally, although rigorously following the PRISMA-type methodology is quite common, there is still the risk of overlooking some related works when conducting a systematic literature review. Therefore, we have also availed several bibliographic resources freely found in the literature.

Finally, to fulfill the last stage (the fifth one), we made a meta-analysis, focusing on PICOT and PICOS, summarizing research questions paradigms. We have also focused on comparatively determining the clinical outcomes with and without rehabilitative kinesiotherapy interventions in Parkinson’s disease patients with COVID-19 infection.

## 3. Results

Initially, 1894 articles were detected. After fulfilling the 5 steps in the selection methodology, including quality evaluation of the articles, 104 papers were selected for synthetic and systematic review. Documentation was enhanced with a supplementary 47 bibliographic resources identified in the literature within a non-standardized search connected to the subject.

Regarding meta-analysis, clinical studies on Parkinson’s disease and COVID-19 were searched on https://clinicaltrials.gov (accessed on 15 January 2022) and were found in 17 such trials. Their analysis for eligibility led to the exclusion of 7 of them, as their content was not limited to this pathology only. The remaining 10 clinical trials presented in Table 1 were used for meta-analysis. We compared the data regarding the main interventions and study type (PICOS): observational or interventional. From a total population of 1196 patients, 759 were included in observational studies and 437 in interventional studies, as presented in Figure 2. An interesting feature we found fulfilling in this research is that all the interventions are rehabilitative kinesiotherapy types, and none of them involved pharmacological interventions. The obtained data can be connected with the finding that, excluding the pandemic conditions, many more clinical trials have physical activities as interventions than other types of pharmacological interventions. A related corollary outcome of our meta-analysis revealed, at least in this very recent period of time (PICOT), as the paradigm and focus of this paper of ours referred only to PD patients infected with SARS-CoV-2, i.e., obviously during this pandemic, it was determined a significant interest in the dedicated clinical trials for the importance of the kinesio-therapeutic approaches.

## 4. Discussion

This article synthesizes the current findings on the pathogenic interference between the dysregulated complex mechanisms involved in aging, the acute and chronic neuroinflammatory repercussions of COVID-19, and neurodegeneration, focussing on PD. Correlations between the putative modus operandi of COVID-19 as a risk factor for neurological infection and its neurodegenerative pathogenic consequences for PD are illustrated in successive steps:Angiotensin-converting enzyme 2 (ACE2) and its borderline role against viral (neuro-) invasion and neurodegeneration;Common central and peripheral nervous structures injured during COVID-19 infection and the morpho-functional dysregulations in PD;Cellular and humoral acute-phase systemic COVID-19 inflammatory mechanisms leading to neuroinflammation-mediated neurodegeneration;Microglia activation as a critical pathogenic mechanism of viral neuroinvasion and self-amplified neurodegenerative mechanisms in PD;Intracellular and molecular pathophysiological interferences between SARS-CoV-2 infection and its pathogenic consequences on neurodegenerative processes in PD:(a)Mitochondria—the primary target for the neuroinflammatory and neurodegenerative etiopathogenesis;(b)Dysregulated proteostasis as a common pathway for COVID-19 and PD;
Self-sustained psycho-neuro-immunological (inflammatory) correlations between COVID-19 and PD;Autonomic dysfunction as the main complaint in long COVID syndrome and PD dysfunctions;Genetics as a backstage for congenital and sporadic forms of PD;Environmental factors and xenobiotics—risk factors deregulating the epigenetic pool in PD;Dysregulation of the individual’s exo- and endogenous bio-ecosystem in aging, PD, and COVID-19;Integrative conventional therapeutic management and non-conventional multitarget approaches to PD and COVID-19 (endogenous engineered factors, exogenous phytonutrients, immunomodulation and immunotherapeutic approaches, nanotechnology, and nanoparticles).

Parkinson’s disease is the second most frequent neurodegenerative age-related progressive condition, significantly increasing in prevalence between 50 and 80 years of age [19]. It is a complex, multifactorial disease affected by diverse genetic, biological, and environmental factors and is assumed to affect 2% of the population over 60 years old. PD is a multifactorial disease induced by a myriad of genetic, biological, and environmental factors. It has a slowly progressive evolutionary trend, clinically characterized by:Motor features (rigidity, bradykinesia, resting tremor, postural instability, impaired balance, and coordination);Nonmotor issues (autonomic dysfunction, such as constipation, urinary dysfunction, postural hypotension);Sometimes cognitive and psychological/neurobehavioral disturbances (depression, psychosis, apathy, sleep disorders);Sensory abnormalities (olfactory dysfunction, paresthesia, pain).The main morpho-pathological hallmarks in PD are:
(a)Progressive and selective degeneration and death of the dopaminergic neurons in the substantia nigra pars compacta (SNpc);(b)Intra-cytoplasmic abnormal aggregation and deposition of a pathologically misfolded protein α-synuclein (aSyn) known as Lewy bodies and other poly-ubiquitinated proteins.

Defective clearance and/or increased levels of abnormal protein aggregates characterize the pathogenesis of NDs [20].

The brains of people with PD present a phenotypic pattern and cellular impairments similar to aging-related mechanisms, comprising impaired functions of the ubiquitin-proteasome system and autophagy-lysosome pathway in the neuronal and glial cellular pool.

Activated microglia turned into a phagocytic morphology represents a similar histopathological hallmark found during senescence and PD [16,18].

Advanced age is a significant risk factor for severe COVID-19 infections among PD patients with age-related comorbidities. However, a systematic review of 13 papers including 928 patients that analyzed the prevalence and prognosis of COVID-19 showed no significant differences between the hospitalization and mortality rate in PD patients versus non-PD subjects, with differences attributed to old age and associated diabetes and immunocompromised status [21].

Other authors noticed a significantly higher COVID-19 incidence in the PD population than in non-PD patients. PD subjects hospitalized for COVID-19 were substantially older males (with mean ages of 80.8 years ± 13.5), severely disabled (HY stage 5), associated with a higher incidence of comorbidity (hypertension and chronic kidney disease), and a higher mortality rate (35.4%) than in non-PD hospitalized patients (67.4 years ± 6.9, respectively, 20.7% deaths) [22].

Dementia of any etiology (Alzheimer’s and Parkinson’s disease) was a prominent risk factor for severe outcomes and death in COVID-19 patients, mainly in the oldest (≥80 years of age) [1,23,24].

Several viral infections, particularly coronaviruses, are assumed to develop NDs due to their direct neurotoxic effects and CNS invasion. They trigger an augmentation of inflammatory markers (D-dimer, ferritin) and proinflammatory “cytokine storm” (IL-1, IL-6, IL-8, TNF), which are capable of disrupting the blood-brain barrier (BBB), inducing microglial activation, brain inflammation, and aSyn aggregation. They finally promote neuronal death by exceeding the molecular and cellular homeostatic mechanisms [9,25,26].

Sometimes post-COVID-19 symptoms can persist beyond the claimed two-week recovery period for mild cases [2,27]. As a result, impairments may extend over 4–12 weeks (up to 6 months) after being tested positive, embedding this category of patients in the ”long COVID syndrome” or “long haulers,” defined by the National Institute for Health and Care Excellence (https://www.nice.org.uk/guidance/NG188 (accessed on 1 February 2021)).

Theoretically, they recovered from the worst impacts of COVID-19 and tested negative, but they still have persistent and disabling symptoms [27].

The pathogenesis of post-COVID-19 complications remains undefined. The severity of an initial viral infection was not a mandatory condition for developing long COVID syndrome in persons living with PD. Paradoxically, the symptoms developed mainly in infected but non-hospitalized patients (77.8%), probably new Parkinsonian symptoms, viral illness-related worsening of pre-existing PD features, secondary to prolonged lockdowns, and reduced access to health care and/or rehabilitation interventions [2].

A multicentric case series across several centers from the United Kingdom, Italy, Romania, and Mexico reported the occurrence of long COVID in 85.2% of hospitalized PD patients. Clinical manifestations were worsening of motor function (51.9%), increased levodopa daily dose requirements (48.2%), fatigue (40.7%), and cognitive disturbances (22.2%), such as “brain fog,” loss of concentration, memory deficits, and sleep disturbances (22.2%) [2].

The acute, subacute, and chronic post-COVID-19 neurological signs and symptoms were attributed to SARS-CoV-2 aggressive neurotropism, triggering a post-viral immune-mediated process.

The presumed pathophysiological mechanisms underlying the long-term neurological sequelae of post-COVID-19 infections are connected to the viral neuroinvasive capacity, hypoxic and inflammatory neuronal injuries, systemic proinflammatory cytokine dysregulation, different degrees of endothelins, BBB injuries, vagus nerve damage, followed by post-COVID-19 dysautonomia, and intracortical GABAergic dysfunctions [26,28,29].

Accumulating data suggest that chronic neuroinflammation-mediated neurodegeneration is considered a significant contributing factor for the neuronal and glial death in the brain during age-related neurodegenerative pathology and might be possibly encountered as a long-term consequence of some viral infections. Similar Parkinsonism-like symptomatology in the elderly was reported post-influenza, SARS, and COVID-19, raising concerns that the successive COVID-19 pandemic waves might induce late chronic neurologic manifestations and new waves of PDs in susceptible patients [28,30,31,32].

The molecular and cellular interferences, clinicopathologic correlations, and the putative modus operandi of COVID-19 that increase susceptibility to neuroinflammation and neurodegeneration, respectively predispose to PD or aggravate it, are schematized in the following pathophysiological items:Intranasal inoculation and/or intestinal gate viral penetration;SARS-CoV-2 infects the cells via the angiotensin-converting enzyme 2 (ACE2) through receptor-mediated endocytosis [10];After coupling to the gate receiver, the virus decreases tissular ACE2 and dysregulates the RAS (renin-angiotensin system) towards the Ang II/AT-1 proinflammatory and pro-oxidative axis [5,33];Viral-induced dysregulation of the ACE2 level exacerbates the severity of infection [5];The endothelial cells have abundant membrane-bound ACE2-type receptors, directly attacked by SARS-CoV-2. Attachment of the viral particles induces endotheliitis and triggers pro-inflammatory, pro-oxidant, and pro-thrombotic mechanisms, contributing to the systemic inflammatory cytokine storm in a vicious cycle [34];Invasion (of the olfactory and/or vagal terminal nerves) is followed by retrograde axonal transport and trans-synaptic transfer directly to the CNS. The hematogenous way is another modality of cerebral dissemination [21];A profuse peripheral systemic immune response, induced by specific cellular protagonists, humoral proinflammatory and neuro-cytotoxic mediators, occurs;Microglia-mediated neuroinflammation is initiated and activated (amplified) by cross-talk between the peripheral inflammatory cells (mast and T cells) after crossing the BBB;The hypoxemia, cerebral hypoxia, neuronal and glial hypoxia induce mitochondrial impairments and an exacerbated oxidative stress;Impaired cellular recycling mechanisms and disrupted proteostasis involve a consecutive accumulation of misfolded aSyn protein (“endocellular junk”), leading to neuronal apoptosis;Influences of the epigenetic factors, xenobiotics, and an unbalanced endogenous microflora (dysbiosis);Genetic factors are backstage predisposing factors, creating a predetermined background.

### 4.1. Angiotensin-Converting Enzyme 2 (ACE2) and Its Borderline Role against Viral (Neuro-) Invasion and Neurodegeneration

ACE2 is a strategic crossroad in the pathogenic associations between the neuroinvasive capacity of COVID-19, neuroinflammation, aging neurodegenerative modifications, and PD.

ACE2 is ubiquitously scattered in the respiratory tract, endothelium, and the cerebral dopaminergic neuronal pool, being a vulnerable gate for viral intrusion, neuroinflammation, and neurodegeneration [10].

The ciliated cells of the nasal mucosa express ACE2 and a transmembrane serine protease 2 (TMPRSS2), which represent primary targets for SARS-CoV-2 infection. The virus reduces ACE-2/angiotensin (1-7)/mas axis activity and its neuroprotective capacity [35,36]. In addition, the spike S1 protein facilitates viral attachment to ACE2 receptors and promotes membrane fusion and endocytosis of the viral particle [37].

Based on the genetic co-expression of ACE2 and dopa-decarboxylase (DDC, the enzyme converting dihydroxyphenylalanine into dopamine and hydroxytryptophan into serotonin), one can assume that a dysfunctional collaboration between ACE2-mediated synthesis of angiotensin 1-7 and disrupted dopamine synthesis might incriminate COVID-19 in triggering PD neurodegenerative mechanisms [33,38].

Dopamine modulates astroglial and microglial activity via the glial RAS (renin-angiotensin system). Reciprocally, cerebral RAS induces dopamine release through modulatory interactions with the dopamine and angiotensin receptors [39].

Neuronal and glial RAS overactivation amplify neuroinflammation and dopaminergic deficiency, revealing the complex intermediate pathways to cellular death [33,40].

### 4.2. The Common Central and Peripheral Nervous Structures Injured during COVID-19 Infection and the Morpho-Functional Dysregulations in PD

The virus spreads from the nasal epithelium, infects the olfactory nerves and olfactory bulb (OB), and then is retrogradely vehiculated through neural pathways to different brain regions (mainly to the brainstem), inducing various neurological deficits: anosmia, hyposmia, ageusia, hypogeusia, other cranial neuropathies, encephalitis or encephalopathy [41,42,43]. Post-mortem studies in subjects who complained of anosmia detected histopathologic abnormalities in the OB [26,44]. MRI studies detected transient edema, degeneration, asymmetric atrophy and reduced volumes (43.5%), scattered hyperintense foci, and microhemorrhages in the OB (91.3%) [43,45].

In most cases, anosmia was reversible after a short time. In severe OB dysfunctions, the impairment was irreversible, reflecting failure in the regeneration attempt of the neuronal pool in the OB and olfactory sensory neurons. Defects in the dopamine system, loss of smell, and OB pathology are characteristic features of PD. Hyposmia is a pathologic hallmark for PD, which progressively worsens. The progressive neurodegenerative loss of smell in PD and post-COVID-19 emergent anosmia might be intercorrelated by the dysregulation of neural stem cells in the OB and olfactory epithelium, with secondary neuroinflammatory lesions and impaired neurogenesis [46].

### 4.3. Cellular and Humoral Acute-Phase Systemic COVID-19 Inflammatory Mechanisms Leading to Neuroinflammation-Mediated Neurodegeneration

Severe SARS-CoV-2 infections are associated with an overactive immune-inflammatory systemic response and the hyperproduction of acute-phase inflammatory cells and mediators.

The p38α MAP kinase plays an essential role in the biosynthesis of proinflammatory cytokines such as interleukin 1β (IL-1β), interleukin-6 (IL-6), tumor necrosis factor-alpha (TNF-α), and the oncogenic activating transcription factor 2 (ATF-2). Activation of p38α kinase was observed in response to SARS-CoV-2 infections, associated with excessive production and release of pro-inflammatory signaling molecules (the cytokine storm) [47]. Higher levels of IL-10 were associated with a more intense innate immune response and headache [29]. Interleukin IL-6 is synthesized by macrophages, T-lymphocytes, endothelial cells, smooth muscle cells, and adipocytes) and stimulates the production and release of hepatic acute-phase protein CRP. Macrophages, adipocytes, and astrocytes are responsible for TNF-α production [48].

The storm of cytokines disrupts the BBB, allowing the peripheral immune cells and inflammatory mediators to induce a pro-inflammatory shift in the nervous parenchyma. High levels of pro-inflammatory markers (IL-6, IL-8, IL-10, and TNF-α) were detected in the cerebrospinal fluid (CSF) collected from COVID-19 patients exhibiting neurological symptoms and/or meningoencephalitis.

An increased plasmatic level of inflammatory biomarkers was detected in all COVID-19 convalescents. High cytokine IL-4 was present at 1 to 3 months in individuals recovering from the initial viral infection. Self-reported persistent neurological problems during COVID-19 recovery were correlated with advanced age, comorbidities, and increased levels of IL6. Post-COVID-19 neurological symptomatology was associated with higher plasmatic levels of SARS-CoV-2 antibodies and elevated plasmatic biomarkers of neuronal dysfunction (amyloid-beta, light neurofilament, neurogranin, tau, and phosphorylated (p-T181-tau)) [16,49,50].

The complex neuro-immunological interrelations and the proinflammatory shift of the brain parenchyma are reflected by the high levels of IL-6, and TNF-α detected in PD patients [48].

Plasma collected from PD patients contains elevated proinflammatory cytokines (IL-6, TNF, IL-1β, and IFNγ). It is still challenging to state whether a neuroinflammatory status has induced PD or this is a consequence of neuronal loss.

The potentially pathogenic relationship between COVID-19 and PD neuroinflammatory mechanisms is based on the virus’ neurotropism and its ability to induce delayed autoimmune mechanisms eliciting neurodegeneration. Frequent detection of anti-COVID-19 antibodies sustains those mechanisms mentioned above, and specific autoantibodies against neuronal or glial epitopes in the CSF were collected from severely ill COVID-19 patients and people with PD [37,51,52].

The persistence of brain inflammatory stimuli, associated with particular genetic and epigenetic risk factors, might promote progressive neurodegenerative modifications and explain long-term neurological sequelae post-COVID-19, including new PD symptoms or worsening of a pre-existing PD [4,6,7,8].

Astroglial toll-like receptors (TLRs) play a crucial role in viral storage, replication, and brain dissemination, acting through the glial pattern recognition receptors (PRRs), which are vital sensors for danger and facilitators of neuroinflammation [53].

Mast cells tryptase acts on the neurons through protease-activated receptor 2 (PAR2), a member of the 7-transmembrane receptor superfamily. The activated receptor PAR2 is synthesized by eosinophils, neutrophils, monocytes, macrophages, dendritic cells, mast cells, and T cells. It modulates the inflammatory responses, acting as a sensor for the proteolytic enzymes generated during infection.

The systemic inflammatory cascade induces microglial activation, demonstrated by the autopsy of COVID-19 subjects [16,50]. Glia and mast cells cross-talk at the brain level, reactivating each other through co-stimulatory molecules CD40/CD154 receptor/ligand or inflammatory mediators such as TNF-α, IL-1β, or IL-33 [54,55].

Microglia-mediated neuroinflammation, in tandem with activated astrocytes, plays an essential role in the central neuroinflammatory process and represents the pathogenic link between the neurotoxic RNA viruses and PD pathogenesis [3,53,56].

T cells, mast cells, reactive astrocytes, and activated microglia are the fundamental “cornerstones” mediating innate immunity during viral neuro infections. They release more proinflammatory mediators in the brain and increase neuroinflammation in a vicious cycle [53].

The proliferation of activated microglia and reactive astrocytes (type A1 phenotype) increases during normal aging. It is abruptly augmented in NDs such as amyotrophic lateral sclerosis, Parkinson’s, Alzheimer’s, and Huntington’s disease. The activated microglia and astrocytes promote a self-sustaining and amplified production of cytokines and create a chronic proinflammatory environment in the brain (a hallmark for both aging and the NDs above) [53].

Excessively activated microglial NLRP3 inflammasomes release proinflammatory cytokines, which are essential “players” in the neurodegenerative pathogenesis and progression, implicitly in PD [57].

An experimental model of PD induced by 1-methyl-4-phenyl-1,2,3,6-tetrahydropyridine (MPTP) confirmed that excessive activation of the microglial NLRP3 inflammasome plays a critical role in the pathological mechanisms of PD [58].

Animal models of PD also demonstrated that defects in microglial autophagy mechanisms could accelerate inflammasome activation and cause PD-like symptoms [59].

Chronic non-resolving inflammation induces NLRP3 inflammasome activation, reflected in mitochondrial dysfunction, a vital organelle for cell energy production [57].

SARS-CoV-2’s pathogenesis (like other coronaviruses) implies the hijacking of the host cell’s biologic machinery for its replication. SARS-CoV-2’s modus operandi aims to divert the physiology of critical organelles and some relevant metabolic pathways.

The virus disturbs proteostasis by attacking Hsp90 (heat shock protein) ubiquitous chaperones, which orchestrate cellular homeostasis, survival, response to stress, hormone signaling, and apoptosis. The heat shock proteins are distributed in the cytoplasm (HSP90 alpha and HSP90 beta), in the endoplasmic reticulum (GRP94 94-kDa glucose-regulated protein), and in the mitochondria (TRAP-1, tumor necrosis factor receptor-associated protein 1). By attacking Hsp90, the virus enhances its RNA polymerase activity and “paralyzes” the host’s cellular proteostasis [60].

Another possible common factor interconnecting the COVID-19 cycle of infection and PD is sorting nexin (SNX27), a protein involved in the endosomal recycling of many important transmembrane receptors.

COVID-19 interferes with host defense mechanisms, subjugates, and monopolizes the endocellular double-membrane vesicles (DMV) generated during the normal autophagy pathway. The virus diverts and exploits DMV as a scaffold for its assembly and replication, facilitating RNA synthesis [61].

Alterations in membrane trafficking pathways have tremendous repercussions on the sorting processes and intracellular cargo manipulation, leading to multiple neurodegenerative disorders [62].

The retromer complex, known as the “master regulator” of endosomal transport routes, directs the traffic of the proteic cargo from the endosome, either via a retrograde pathway to the Golgi network or through a recycling pathway back to the cell surface. The vesicular endosomal system is a strategic target for COVID-19 proliferation/invasion. SARS-CoV hijacks the host`s endoplasmic reticulum (ER) and retromer complex, using them to synthesize and process viral proteins. [63].

The spike S protein enters and accumulates inside the ER and activates a cellular signaling pathway known as the unfolded protein response (UPR). The upregulated UPR represents one of the viral strategies to combat cellular response and facilitate viral replication [64].

With the inhibition of the ubiquitin-proteasome system, UPR amplifies the accumulation of dysfunctional misfolded/unfolded proteins [36]. Mutations in the retromer’s structure (occurring in sporadic and atypical PD) can disrupt the endosomal transport routes, promote unfolded protein response aggregation, and accumulation of “junk” aSyn misfolded proteins [36,63].

COVID-19 strategically subverts the double-membrane vesicles/vacuoles generated during the autophagy cascade (the autophagosomes and phagolysosomes), deceiving them for transcription, replication, translation, and/or viral release. Some RNA viruses can successfully activate autophagosome initiation to achieve a maximum accumulation of autophagic vacuoles and use them for viral replication. The viruses block the organelle maturation towards phagolysosomes and inhibit the clearance of the pathogen agent [65]. By interfering and disrupting the autophagosome-lysosomes, COVID-19 facilitates abnormal proteic aggregation and intraneuronal “metabolic pollution,” possibly promoting similar long-term metabolic disruption as in NDs, such as the Lewy bodies in PD [66].

Parkinsonism-like symptomatology reported in the elderly as a late viral manifestation of post-influenza or COVID-19 might probably be determined by aSyn accumulation and cross-autoimmunity reactions elicited by the viral infections [28].

Analyzing the complex links between the pathogenic cascades of cellular-molecular events triggered by COVID-19, one can assume that viral neurological infections might trigger risk for future Parkinsonism-like symptomatology and persistent neurologic manifestations in susceptible patients [30,31,32].

However, after two years and five successive waves of pandemic evolution, just a few case reports described clinical signs of Parkinsonism after COVID-19 and a spontaneous improvement in one subject [32,67,68]. The exact mechanisms of chronic neuroinflammation in PD remain elusive [15,57].

### 4.4. Microglia Activation as a Critical Pathogenic Mechanism of Viral Neuroinvasion and Self-Amplified Neurodegenerative Mechanisms in PD

Neuroinflammation-mediated neurodegeneration is an important etiopathogenic mechanism contributing to neuronal death in the brain of people with age-related NDs.

SARS-CoV-2 infection might be the crossroad between aging-specific dysfunctional metabolic decline, neurodegeneration, and worsening of pre-existing chronic conditions [36].

During the pandemic, different genomic alterations generated about 17 variants of protein-membrane ORF3a (open reading frame 3a—accessory protein), which promoted viral replication [69].

ORF3 occupies a nodal place in prompting neuroimmune phenomena and maintaining neurodegenerative processes. ORF3 is a potent activator of the pro–IL-1β gene transcription and protein maturation required to activate the NLRP3 inflammasome and microglial activation [70].

The two-signal model for inflammasome activation is realized by a functional tandem ORF3a-TRAF3 (tumor necrosis factor receptor-associated factor 3):At the nucleus level, it stimulates pro–IL-1β gene transcription and synthesis of the cytokine precursor (pro–IL-1β) [70];In the cytoplasm, they induce caspase-1 to realize proteolytic activation of the proinflammatory cytokine IL-1β, resulting in cellular death via pyroptosis (caspase 1-dependent cell death), a crucial process in controlling infections [71].

The viral ORF-9b also localizes in the mitochondria and disrupts its primary functions. It fails the redox homeostasis and energy shortage in the infra-microscopic cellular universe and finally leads to the disintegration of organelles and autophagy of the host cell [6,36,46,72].

Inflammasomes are cytosolic polymer complexes that coordinate the host’s innate immune defense mechanism against exposure to pathogens and cellular damage [73]. These cytosolic proteinaceous complexes recognize many perturbing stimuli, such as accumulating danger-associated molecular patterns (DAMPs) and pathogen-associated molecular patterns (PAMPs), which trigger the assembly of NLRP3 inflammasomes. NLRP3 inflammasomes are essential components of the innate immune system and are activated by multiple molecular and cellular stimulation (ionic flux, mitochondrial dysfunction with excess ROS, and lysosomal damage) [55]. Abnormally aggregated aSync induces NLRP3 inflammasome activation [56,73].

Activated inflammasomes lose their autoinhibited steady-state and initiate pro-caspase-1 self-cleavage to the active caspase-1, which in turn triggers glial maturation and secretion of proinflammatory cytokines IL-1β and IL-18, TNF, as well as pyroptotic cell death [73,74,75]

Microglial NLRP3 inflammasome activation has a critical role in the mouse model of PD induced by 1-methyl-4-phenyl-1,2,3,6-tetrahydropyridine (MPTP) [58].

Inflammasomes can be deleterious in chronic brain inflammatory states, and their dysregulation leads to neuronal pyroptosis and contributes to progressive chronic neurodegeneration by accelerating microglial activation [70,71].

Activation of the inflammasome is not exclusively a brain event. Systemic activation of NLRP3 inflammasomes was detected in the peripheral blood mononuclear cells, along with high levels of the inflammatory cytokine IL-1β and an elevated plasma amount of aSyn (at higher levels than control). The systemic activated inflammasomes in the mononuclear cells and aSyn levels in plasma are correlated with motor disability and progression in PD. Tracking these biological markers might be a valuable tool for monitoring the outcomes and progress of PD [75].

### 4.5. Intracellular and Molecular Pathophysiological Interferences between SARS-CoV-2 Infection and Its Pathogenic Consequences on Neurodegenerative Processes in PD

#### 4.5.1. Mitochondria—The Primary Target for the Neuroinflammatory and Neurodegenerative Etiopathogenesis

Mitochondria are the “cellular energetic central”, dynamic organelles vital for energy production. Mitochondrial oxidative phosphorylation is the key to the energy-producing process in the brain.

The mitochondrial respiratory chain complexes I–IV support the electron transport system and establish a proton gradient across the inner mitochondrial membrane [76]. Impairments of the mitochondrial bioenergetic pathway and subsequent mitochondrial redox imbalance might be the alleged etiopathology clue to systemic disorders, resulting in a disruption of proteostasis, dysregulation of neuromediators, neuronal death, multifactorial NDs (amyotrophic lateral sclerosis, Parkinson’s, Alzheimer’s, and Huntington’s diseases), and other pathological conditions (age-related macular degeneration, and glaucoma) [19,77]. Mitochondriopathies are linked to the aging neurodegeneration process [76,77,78].

Oxidative stress involves the chemistry of so-called reactive oxygen species (ROS) derived from oxygen and nitrogen, such as peroxides, superoxide, hydroxyl radical, and singlet oxygen [79].

Mitochondrial physiology is mediated by PINK1 (a ubiquitin kinase) and Parkin (an E3 ubiquitin ligase). They cooperate in fine-tuning control, removing damaged and dysfunctional organelles via mitophagy (a selective form of autophagy). PINK1 accumulates on the outer membrane of damaged mitochondria and activates Parkin E3 ubiquitin ligase to “tag” the dysfunctional mitochondria.

Parkin plays a crucial role in the clearance of ROS and mitophagy. Parkin ubiquitinates proteins on the outer mitochondrial membrane and recruits the autophagy receptors to trigger selective mitophagy [80].

DJ1 belongs to the peptidase C56 family, a molecular chaperone with protease activity. It is a redox sensor and antioxidant scavenger, protecting mitochondria against oxidative stress and maintaining their energetic homeostasis. It protects the dopaminergic neurons against neurodegeneration in PD [77,80].

COVID-19 s responsible for severe acute respiratory insufficiency/ distress, subsequent hypoxemia, sepsis-induced hyper-coagulation, multisystemic injuries, local thrombosis in brain vessels with hypoperfusion, and cerebral hypoxia. COVID-19 exacerbates oxidative stress dysregulation of the redox homeostasis at the cellular level and promotes neurodegenerative modifications.

COVID-19 viruses inhibit immune signaling by targeting mitochondrial-associated antiviral signaling (MAVS) [81] to escape the host’s innate immunity.

Viral ORF-9b localizes to mitochondria and disrupts its essential functions, produces failure of the redox homeostasis, provokes a general energy shortage in the infra-microscopic cellular universe, impairs mitochondrial proteostasis, and finally, leads to the disintegration of organelles and autophagy of the host’s cells [6,36,46].

Mitochondrial degradation in patients with COVID-19 is associated with the intra-and extracellular release of mitochondrial DNA (mtDNA). Circulating cell-free mitochondrial DNA (Cf-mtDNA) consists of mtDNA fragments released outside the cell, then into the circulation by cell necrosis and secretion [82,83].

These circulating mitochondrial fragments are an early indicator of severe viral illness and high mortality risk and are positively correlated with the circulating pro-inflammatory cytokines IL-6, MIG, MCP-1, IP-10, IL-1RA, IL-2R, and HGF [83,84].

In a vicious cycle, Cf-mtDNA amplifies inflammation and aggravates tissue injuries by activating toll-like receptor 9, inflammasomes, and the stimulator of the interferon gene pathway [82,85].

PD has a multifactorial etiopathogenic background. Dysfunctions of the mitochondrial and lysosomal systems are involved in neurodegenerative transformations occurring in PD.

Mitochondrial issues in PD are associated with dysfunctions of the energetic complex I (the main entry point of the mitochondrial respiratory chain), prompting excessive oxidative stress and elevated levels of ROS, impaired iron metabolism, and an increased dopamine turnover misfolded aggregation of aSyn, and neurotoxicity [76]. An excessive number of dysfunctional mitochondria are responsible for early-onset PD pathogenesis [77].

The antioxidant glutathione clears excessive oxidative stress and elevated levels of ROS. A reduced glutathione level is coupled with a high dopamine turnover, overproduction of ROS, and the subsequent cytotoxic cascade of lipid peroxidation, leading to neuronal death.

Dopamine is metabolized by the monoamine oxidase (MAO)-B, generating hydrogen peroxide and dopamine-quinone (a byproduct resulting from dopamine oxidation).

Accumulation of oxidized dopamine-derived residues entails a time-dependent pathological degenerative cascade:Structural and conformational changes of proteins;Mitochondrial oxidative stress;Reduced glucocerebrosidase enzymatic activity;Parkin, DJ-1, and ubiquitin C-terminal hydrolase L1 (UCH-L1) lysosomal dysfunction;Nonselective protein modifications and aSyn accumulation (pathophysiological markers for PD) [35].

Oxidative damage is the plausible mechanism generating pathological aggregation of aSyn, promoting dopaminergic neurotoxicity and PD [57,86]. The accumulation of cytosolic aSyn aggregates in neurons impairs mitochondrial function, inducing the generation of mtDNA and mtROS [57]. The abnormal aSyn “junk” proteic aggregates activate the NLRP3 inflammasomes in microglia through interaction with toll-like receptors (TLRs) [57].

These TLRs are a family of pattern recognition receptors (PRRs) that sense pathogen-derived (PAMP) or endogenous ligands (DAMP) released by damaged cells and initiate the innate immune response. Ten TLRs have been identified in humans. TLR2 and TLR4 are bridging the autoimmune pathology and the neurodegenerative mechanisms. Both TLR2 and TLR4 play a role in PD pathogenesis, inducing microglial activation [87]. Even in the absence of aSyn pathological aggregates, microglial inflammasomes can be activated by αSyn fibrillar and dopaminergic degeneration and generate a cerebral source of neuroinflammation and sustained progressive neurodegeneration [57,88]. The interplay between the progressive aSyn metabolic dysregulation (synthesis, degradation, abnormal storage, extracellular expulsion) and the chronic NLRP3 inflammasome activation is still poorly understood [88].

The consequences of mitochondrial abnormalities and energy shortage are damaged cellular proteostasis, lysosomal impairments, dysfunctions of the autophagy mechanisms, and aspects involved in aging and NDs.

#### 4.5.2. Dysregulated Proteostasis—Common Pathways for COVID-19 and PD

The mainstay hallmark of NDs is the intra- and extracellular accumulation of metabolic waste products [36]. Alpha-synuclein is mainly a neuronal protein and is also detected in the glial cells [89]. It is found predominantly in the presynaptic termini and is highly expressed in the mitochondria and cytosol of the striatum, OB, hippocampus, and thalamus. In contrast, neocortical and cerebellar neuronal pools contain rich cytosolic quantities of aSync but low levels of mitochondrial aSyn. This asymmetric distribution correlates with clinical symptoms and the evolution of PD. Alpha-synuclein controls the mitochondrial function under both physiological and pathological conditions [19].

Under physiological conditions, the monomers of aSyn participate in dopamine exocytosis/neurotransmission and recycling, acting as a molecular chaperone that binds to VAMP2 (synaptobrevin) and stabilizes SNARE-dependent vesicle synaptic fusion complexes (soluble NSF attachment protein receptors) [90].

In cellular pathologic stress conditions (oxidative stress, mitochondrial impairments, proteasomal and lysosomal dysfunctions), secretion of aSyn is significantly accentuated. It accumulates and aggregates, resulting in insoluble intracellular inclusions [91,92]. Pathological aSyn aggregates in mitochondria inhibit in a dose-dependent manner the complex I activity of the mitochondrial respiratory chain in the aforementioned predisposed topographic neuronal areas, leading to neurodegeneration [19].

The accumulation of misfolded aggregated pathological proteins and the particular topographic regions in the brain containing these proteic “hallmarks” are pathogenic characteristics of different synucleinopathies, including PD, dementia with Lewy bodies, and multiple system atrophy.

These pathologically polymerized intraneuronal aSyn aggregates represent the major component of Lewy bodies (LB) and Lewy neurites (LN). They are responsible for dopaminergic neuron death and subsequent loss of motor control (at a certain critical level of neuronal destruction).

Intracytoplasmic accumulation of aSyn aggregates links the pathogenic mechanisms of neurodegeneration to endoplasmic reticulum (ER) stress. The unfolded protein response (UPR) is a cellular mechanism linked to ER pathological stress and used by COVID-19 to defeat the cellular combat response and facilitate viral replication. Different pathological situations disrupt UPR, causing an increased accumulation of unfolded proteinaceous aggregates inside the ER [46,93].

Neurons and glial cells have a sophisticated system for the quality control and maintenance of proteostasis, aimed at detecting and removing aberrant proteins and preventing detrimental aggregation of “junk” misfolded proteins at an early stage.

Dysfunctions in autophagic and/or lysosomal-related processes lead to the pathological accumulation of protein aggregates and represent central pathogenic mechanisms underlying brain neuroinflammatory and neurodegenerative disorders [65,92]. Both neurons and glia are “polluted” by these insoluble “junk-protein deposits”. The intracellular toxic accumulation of aggregated aSyn disturbs the molecular mechanisms responsible for the cellular “housekeeping,” such as the ubiquitin-proteasome system (UPS) and autophagy-lysosome pathway (ALP), leading to neuronal death [65].

The autophagy mechanisms and vesicular trafficking are decisive for the neuronal maintenance of axon homeostasis and recycling the synaptic proteins [65].

Chaperone-mediated autophagy (CMA) is a lysosomal-dependent protein degradation pathway. The two major “players” of CMA are LAMP-2A and HSC70. Pathological aSyn aggregates can resist degradation and impair the CMA and macroautophagy [92].

The most common cause of familial PD dysfunctional autophagy is associated with the expression of dominant mutants of LRRK2 (leucine-rich repeat kinase 2). LRRK2 mutations disrupt both DA neurons as well as the astrocytes.

LRRK2-mutant dopaminergic neurons have an altered CMA process, leading to abnormal aSyn accumulation, preceding accumulation of LB [37,94].

Transcription factor EB (TFEB) is a master gene for lysosomal biogenesis and autophagy and encodes a transcription factor to coordinate the synthesis of lysosomal hydrolases [92]. It can induce intracellular clearance of pathogenic protein markers and regulate lysosomal dyshomeostasis [20,95].

Proteiform aggregates that overwhelm the cellular clearance capacity are released in the extracellular space between neurons and glia by unconventional exocytosis. These toxic forms of aSyn are transported via extracellular vesicles in the biological fluids (CSF, plasma, salivary fluids, and urine) [16,91,96,97,98].

The exosomes released from dopaminergic cells contain PD-related proteins (aSyn, LRRK2, and DJ-1) carried in double-membrane extracellular vesicles (EVs). The protein cargo in the EVs is used as a laboratory biomarker, reflecting the outcomes of PD evolution [96,97,98].

The exosomes and double-membrane extracellular vesicles are found in the Meissner’s and Auerbach’s plexuses of the gastrointestinal tract in the developing evolutionary stages of PD. This could support the two-way trafficking of nanoparticles filled with pathological PD-related proteiform aggregates, transported downward through the vagal neural connections (although the peripheral deposits might accumulate by systemic circulation) [91,99].

Toxic aSyn protein aggregates can propagate from cell to cell, between neurons and glia, interacting with other local proteins in a prion-like fashion, performing as a matrix or a nucleating seed and promoting new pathological conformational changes and new LB aggregates. The pathologic form of aSyn extends gradually in an “oil stain” manner, affecting non-dopaminergic neurons in other brain regions and being responsible for non-motor symptoms in PD.

Extracellular aSyn released from the dopaminergic neurons represents an endogenous agonist for toll-like receptor 2 (TLR2), linking innate and adaptive immunity [100]. These sophisticated TLRs receptors mediate the clearance of the extracellular aggregates and neuroinflammation involved in the initiation and progression of different NDs [100].

### 4.6. Self-Sustained Psycho-Neuro-Immunological (Inflammatory) Correlations between COVID-19 and PD

COVID-19 induced heterogeneous clinical manifestations, multisystemic organ damage, and complications including neurological, psychiatric, psychological, and psychosocial impairments.

The cognitive and psychological/neurobehavioral disturbances (depression, psychosis, apathy, sleep disorders) are part of the clinical picture of PD disease.

PD patients have multiple patterns of vulnerability associated with age-related chronic comorbidities and polypharmacy burden.

The pandemic was a significant stressor worldwide, and PD patients experienced psychiatric and psychological disturbances. They felt high levels of chronic stress and depression due to the imposed lockdown measures, fear of being infected and death, and lack of a systematic physical kinetic program, all converging to worsening motor and non-motor symptoms.

During long-time exposure to chronic stress and COVID-19 infection, the systemic acute-phase proteins and proinflammatory cytokines (IL-1, IL-6 TNF-α) represent modulators of the hypothalamic-pituitary-adrenal endocrine axis and stimulate cortisol secretion [48,74,101].

The psycho-neuro-immunological/inflammatory repercussions might trigger a self-sustaining response in a vicious cycle, activating the inflammatory response system and increasing serotonin and catecholamine turnover [48,101].

### 4.7. Autonomic Dysfunction, the Main Complaint in Long COVID Syndrome and PD Dysfunctions

Autonomic dysfunction is a possible long-term complication in post-COVID patients, with or without neurological symptoms or complications. The most frequent issues are orthostatic hypotension, postural tachycardia syndrome, secretomotor, urinary, and pupillomotor impairments [97,102].

A myriad of pathophysiological interferences might concur with the pathogenesis of dysautonomias in a perpetual vicious cycle. Viral interference with the ubiquitous ACE2 (the primary cell entry receptor for the SARS-CoV-2), different degrees of endotheliitis, inappropriate immune responses, damage of the enteric vagal visceral afferents to the brain and hypothalamic-pituitary-adrenal axis, and dysbiosis interferes with post-COVID-19 dysautonomia [28,34,103].

Orthostatic hypotension and postural tachycardia post-SARS-CoV-2 infection can be commonly encountered in the elderly and in individuals with peripheral neuropathies and/or dysautonomic nervous pathologies such as PD, dementia, or diabetes [102].

### 4.8. Genetics as a Backstage for Congenital and Sporadic Forms of PD

Most PD cases are not hereditary but idiopathic (with unknown etiology). Identical twin studies have suggested that something else overlaps with a predisposing genetic dowry, inducing PD.

The complex pathogenesis of PD results from combinations of inappropriate lifestyle and toxic environmental factors in interactions with variants of PD-related genes transmitted with non-Mendelian inheritance, acting as predisposing genetic factors [104,105,106]. Genetic variants are implicated in about 15% of cases that occur in people under 50 years and are known as early-onset PD [91,107].

The frequency of different gene variants has great variability in populations. Mutations of more than 20 genes (most of them highly penetrating) encode an early-onset phenotype of PD [106]. Mutations induce family-linked PD cases in the mtDNA or nuclear DNA of the following genes: E3 ubiquitin ligase (Parkin), aSyn, ubiquitin carboxy-terminal hydrolase L1 (*UCHL1*), a Parkin-associated protein involved with oxidative stress (*DJ1*), putative serine-threonine kinase (*PINK1*), auxilin (*DNAJC6*), synaptojanin 1 (*SYNJ1*), serine peptidase 2 (*HTRA2)* and endophilin A1 (*SH3GL2*). These are responsible for PD [77,91,94,107].

Polymorphisms of *DNAJC6*, *SYNJ1*, and *SH3GL2* are associated with disruption of endocytosis and mitochondrial function and are related to neurodegenerative pathogenesis of PD [77].

Mutations in the aforementioned genetic pool contribute to the development of PD and associated neurodegeneration at different cellular-molecular levels: dysfunctions of the membrane compartments, impairment of the mitochondrial system and energetic failure, enzymatic loss-of-function of the autophagy-lysosomal processing and clearance and substrate accumulation, which lead to proteostasis alteration and misfolded aggregates of aSyn (eventually associated with pathological tau protein), inducing metabolic pathway alterations in a bidirectional feedback loop.

Family-linked cases involving PARK7, PINK1, or PRKN genes determine the occurrence of autosomal recessive inheritance patterns. Disorders of LRRK2 or SNCA genes induce an autosomal dominant pattern [77,91,94,107].

LRRK2 (leucine-rich repeat kinase 2) is a multifunctional kinase expressed in human neurons, glial cells, and other tissues throughout the body responsible for delivering phosphates to other proteins. Dysfunctional LRRK2 provokes the accumulation of phosphorylated aSyn, altering cellular functions and signaling pathways.

Mutations of the LRRK2 gene are the culprit for neurodegeneration in PD. Patients with LRRK2-associated PD represent the most common causes of autosomal dominant inherited PD (1–2%, up to 3 and 4% of PD subjects worldwide) [77,91,94,107].

Mutant LRRK2 inhibits CMA complexes and provokes an accumulation of toxic intracellular aSyn aggregates in the dopaminergic neurons [94,108]. The LRRK2 gene upregulates the transcriptional activity of NF-kB (nuclear factor-kappa light chain enhancer of activated B cells), a family of transcription factors mastering the innate inflammatory immune responses and apoptosis [108].

Mitochondrial physiology is mediated by PINK1 (a ubiquitin kinase) and Parkin (an E3 ubiquitin ligase). Genetic mutations of the signaling couple PINK1/Parkin are associated with early-onset familial forms of PD. Defective structural mitochondria and dysregulated genetic byproducts (PINK1 and/or Parkin) mediate neurodegeneration and neuroinflammation by disturbing ROS clearance and selective mitophagy, impairing the energetic cellular metabolism [109]. A dysfunctional ubiquitin system (Parkin) can trigger NLRP3 inflammasome activation [71,73].

DJ1 is an integral mitochondrial protein that regulates mitochondrial homeostasis and maintains the functional activity of the mitochondrial complex I (the dysfunctional key point in PD) [77].

HTRA2 is also a mitochondrial protein responsible for the quality control and homeostasis of the mitochondrial intermembrane space. Mutations in HTRA2 are associated with autosomal dominant late-onset PD [110].

UCHL1 (ubiquitin carboxy-terminal hydrolase L1) is a key enzyme for cellular proteostasis. Disturbances in the protein degradation pathway contribute to energetic mitochondrial failure and synaptic dysfunctions [77].

The SNCA gene causes metabolic disruptions of the aSyn protein, inducing its oligomerization and fibrillization. Alterations of the lysosome-related genes impair the degradation of aSyn misfolded proteins and induce neurodegenerative modifications.

Depleting the ATPase cation that transports 13A2 (ATP13A2) leads to aSyn accumulation through lysosomal dysfunction. Mutations in lysosomal hydrolases (i.e., mutated glucocerebrosidase) play a critical role in PD [20,107]. The hereditary PD-phenotype clinically and pathophysiologically resembles the idiopathic forms of the disease [94].

### 4.9. Environmental Factors and Xenobiotics—Risk Factors for Deregulating the Epigenetic Dowry in PD

Environmental factors play a major role in human chronic diseases. The exposome concept refers to various exogen and/or endogen exposures, including xenobiotics (neurotoxic chemical agents, biological agents, or radiation), over a complete lifetime.

The etiopathogeny of PD may imply neurodegenerative epigenetic/environmental factors that are not entirely understood. The xenobiotics might impair essential cellular organelles involved in autophagy and proteostasis. Exposure to toxins such as rotenone, paraquat, MPTP, and a supranutritional intake of selenium is briefly pointed out [107,111].

Ingestion or long-term exposure to the herbicide paraquat is associated with acute or chronic pulmonary inflammation, progressive lung fibrosis, liver and kidney lesions, and PD. Altered mitochondrial oxidative stress is responsible for its toxicity [79,112].

Bacterial or viral toxins, some mineral crystals, lipids, β-amyloid, and L-leucyl-L-leucine methyl ester (LLOMe—a lysosomotropic agent) might induce neurodegeneration of dopaminergic neurons [66].

Systemic exposure to MPTP (1-methyl-4-phenyl-1,2,3,6-tetrahydropyridine), a toxic contaminant of street drugs (illegally produced in underground laboratories) causes Parkinsonism in drug abusers. https://www.cdc.gov/mmwr/preview/mmwrhtml/00000360.htm (accessed on 15 January 2022).

MPTP is lipid-soluble, penetrates the BBB, and is captured into astrocytes, where it is then converted to MPP+ by monoamine oxidase-B (MAO-B). The dopaminergic neurons selectively take up MPP+, resulting in neurotoxic effects via GSK-3 (glycogen synthase kinase). Pathological amounts of GSK-3 trigger an excessive aSyn and tau phosphorylation, leading to neurotoxic aggregates, loss of dopaminergic neurons, and neurodegeneration [107,113,114].

### 4.10. Dysregulation of the Individual’s Exo-/Endogenous Bio-Ecosystem in Aging, PD and COVID-19

Paradoxically, more than half of our body is not human. Human cells make up only 43% of the body’s total cells; the rest belongs to the endogenous and exogenous “microscopic galaxies” (bacteria, viruses, fungi, and archaea—organisms initially misclassified as bacteria) co-existing with the human body. The human genome consists of 20,000 genes (“human DNA”). The number of genes in our microbiome is estimated to be between 2 and 20 million genes (mainly the DNA of our gut microbes). https://www.bbc.com/news/health-43674270 (accessed on 15 January 2022).

Only a tiny minority of the human genome (barely 2% of the total genetic pool) encodes structural proteins. The “cosmic” majority of the human genome is not encoded for morpho-functional proteic purposes. These non-coding genes are transcribed from DNA to functional RNA molecules (ncRNA) but not translated into proteins. They control gene expression at the transcriptional and post-transcriptional levels (supervising the “fine-tuning” in the cellular universe, regulating the expression of protein-coding genes) [115].

Environmental bacteria co-evolved in symbiotic coexistence, inside and outside the human body, and modulated the activity of our entire organism. The greatest concentration of these microorganism colonies is located in the “dark murky depths” of our oxygen-deprived bowels. Based on the fact that there are 90 billion bacteria in 1 g of feces, the total number of bacteria colonizing the intestines is estimated at 38 trillion.

This hidden half of “ourselves”—our endogenous microbiome—is “deeply” implied in the etiopathogenic of progressively neurodegenerative disorders. Our endogenous microbiome influences the brain through the brain-gut-microbiota axis, and communication dysregulations are involved in aging-related neurodegenerative impairments [116,117].

Clinical pathophysiological data sustain the hypothesis that dysfunctions induce the onset of PD in the gut microbiota, retrogradely transmitted through trans-synaptic connections from cell to cell via the ascending gut-brain axis of the sympathetic and parasympathetic nervous system [118].

As part of our exposome, the microbial or viral illnesses appearing earlier in life (as part of our exposome) might be responsible for late-onset synucleinopathies [119].

Age-associated gut dysbiosis (abundance, composition, and diversity) in the elderly is characterized by an enrichment of proinflammatory microorganisms to the detriment of beneficial strains. This shift induces a pathological and enhanced intestinal permeability, systemic inflammation with disruption of the BBB selective permeability, then neuroinflammation, alterations of the brain metabolites, neuromodulators, and neurotransmitter precursors, and impaired synaptic plasticity [103].

The altered quality of gut microbiota represents a predisposing background for COVID-19 and reveals the pivotal role of dysbiosis in the high fatality rate of COVID-19 in elderly patients [103,120].

Probiotics administered in COVID-19 aimed to modulate the exacerbated immune responses and inhibit the secretion of proinflammatory cytokines [118].

PD patients have an altered gut microbiota, with significant variations in bacterial strains, some being upregulated and some downregulated [117,118].

Molecular similarities between food antigens and food-specific antibodies in the blood might induce cross-reactivity to aSyn and predispose to neuroinflammation and synucleinopathies [121].

Gliadins and glutenins, the main components of the gluten fraction in wheat, are responsible for celiac disease. The 33-mer gliadin molecule has significant biochemical homology with the GRINA component of the human NMDA glutamate receptor (GRINA, glutamate ionotropic receptor NMDA type subunit associated protein 1). Dietary imbalance and food-specific antibodies impair glutamate receptors and produce dysfunctions in neurotransmission, playing an essential role in PD pathophysiology [121].

Exposure to SARS-CoV-2 in the above-mentioned epigenetic interferences might represent a trigger risk for neurodegenerative processes and the future development of Parkinsonism-related symptoms.

### 4.11. Integrative Conventional Therapeutic Management and Non-Conventional Multitarget Approach of PD and COVID-19 (Endogenous Engineered Factors, Exogenous Phytonutrients, Immunomodulation, and Immunotherapeutic Methods, Nanotechnology, and Nanoparticles)

Over 200 years since the first description of PD, published in 1817, there is no curative treatment for PD. Currently, available therapies focus only on symptom management and are associated with incapacitating adverse effects.

People with PD have heterogeneous symptoms and evolutive trends, demanding adapted health management strategies.

Treatment is symptomatic and centered on improving the motor and nonmotor symptoms. Actual pharmacotherapy aims at the compensation for dopaminergic deficiency and symptomatic treatment.

A tailored and integrative treatment strategy for PD is the usual protocol, with the “classical” treatment options: rehabilitation, neurorestorative endeavors, and surgery. The ultimate goal of a management program is to improve the quality of life for both the patient and his caretaker [122].

Drugs targeting debilitating PD motor symptomatology include levodopa (the gold standard in PD treatment), dopamine agonists (apomorphine, ropinirole, and pramipexole), monoamine oxidase inhibitors (selegiline and rasagiline), anticholinergic drugs (trihexyphenidyl), and catechol-O-methyl trans (entacapone). Symptomatic therapy may include antipsychotics or antidepressants.

Besides their positive effects, the current pharmacotherapeutic approach may be accompanied by adverse reactions such as dyskinesia or fluctuating therapeutic effects.

The ultimate therapeutic goal (to stop and/or reverse the progression of PD) remains elusive, and new steps to achieve an individualized and holistic treatment are continuously sought [123].

Amantadine was initially used as an antiviral agent, then beneficial pharmacological effects were reported in patients with PD. The molecular mechanism of PD is not fully understood.

In confirmed COVID-19 patients suffering from PD, a daily dose of 100 mg amantadine or twice-a-day 10 mg memantine demonstrated beneficial effects on both PD and COVID-19.

There are no relevant randomized clinical trials to address the neuroprotective effect of amantadine administered in SARS-CoV-2-infected PD patients [124].

Apomorphine is the most potent dopaminergic agonist, but it cannot be assimilated through the digestive tract. It must be injected under the skin using a computerized pump. Besides this usual route, intranasal, sublingual, rectal, and iontophoretic transdermal delivery routes have been investigated.

Contemporary research developed the spectrum of delivery, creating a thin film of apomorphine and a levodopa inhaler, both approved in 2020 by the US FDA [125].

A review summarizing 12 years of clinical experience revealed that a transdermal rotigotine patch could alleviate motor issues in PD and improve sleep disturbances, neuropsychiatric symptoms, and health-related quality of life. Due to its safety and tolerability profile, the patch was used as an alternative to conventional oral treatment in PD. Its efficiency was proved in nil-by-mouth scenarios encountered in an emergency, intensive care, or a healthcare crisis, such as the current pandemic [126].

Considering a positive diagnosis in PD is made around age 60, and many patients survive between 10 and 20 years, health management strategies must be tailored for different clinical stages of the disability in a geriatric context of co-morbidities.

The FDA has considered some anti-aging molecules such as metformin, rapamycin, and resveratrol to reduce cellular damage and neurodegeneration and prolong the health span [18].

Deep brain stimulation (DBS) has been the most important therapeutic advancement in PD since levodopa development [127].

A recent pilot trial study referring to neuromodulatory electrostimulation of the cuneiform nucleus indicated in levodopa-resistant freezing of gait was interrupted due to the global COVID-19 context [128].

Physical therapy is an essential component of the standard package of therapeutic interventions. During the pandemic, prolonged homestay due to lockdowns reduced access to health care and rehabilitation interventions. It accentuated the clinical manifestations resulting from new additional PD symptoms, lockdown-induced anxiety, and depression, as well as the viral-related worsening of pre-existing PD features [2].

Telerehabilitation offered high-quality care programs and an optimal method to continue kinesiotherapy in areas without accessible rehabilitation clinics or during lockdown periods [129,130].

The hybrid model, associating the standard in-person medical assistance with remote-delivered healthcare as an alternative to conventional face-to-face physiotherapy, might represent a flexible rehabilitation model during the pandemic [131].

The modern individualized, holistic management of chronic neurodegenerative conditions targets the development of non-conventional multitarget strategies to limit brain neuroimmune and neurodegenerative pathology and promote neuroregeneration.
Endogenous engineered factors:Erythropoietin (EPO) can restore cell viability and contribute to dopaminergic neurorestorative mechanisms. in PD rodent models, administration of recombinant human erythropoietin (Rh-EPO) or EPO analogs revealed neuroprotective and curative effects against MPTP-induced PD. In humans, two pilot studies investigated the safety and efficacy of rhEPO. They found it a potentially useful agent against oxidative stress, restoring the redox imbalance and neuroinflammation associated with PD [78,124];Lactoferrin (Lf) is an iron-binding glycoprotein involved in oncology immunomodulation, neuroprotection, and apoptotic processes. Lf can combat COVID-19-related inflammation by acting as a natural barrier at the respiratory and intestinal mucosa levels, reversing iron disorders related to the viral invasion, and down-regulating the proinflammatory cytokines and a generalized cytokine storm [132]. Considering its iron-binding capacity, Lf has a neuroprotective effect, preventing dopaminergic neurons’ spontaneous and progressive death and being indirectly beneficial to microglial cells. Lf is synthesized by activated microglia and binds to heparan sulfate proteoglycans (HSPGs), acquiring a protective effect for SN neurons [132];Mesenchymal stem cells (MSCs) are considered the cornerstone in a galaxy of intercellular signals. They represent an important endogenous neuroprotective and immunomodulatory factor. Clinical trials demonstrated the therapeutic efficacy of MSCs in patients with COVID-19 through their anti-inflammatory effects against cytokine storms, being an alternative option for critically ill patients with COVID-19 [133];
Exogenous phytonutrients (the “nutrients found in a plant”) are natural phytochemical products with diverse pharmacological activities, which exert a robust anti-oxidant action and represent promising therapeutic tools against the complex pathophysiological disorders underlying central manifestations of COVID-19 [134];Lemon IntegroPectin was obtained from (organic) lemon processing waste using hydrodynamic cavitation. It exerts a potent antioxidant activity, a significant mitochondrial protective action, and neuroprotection, demonstrated in vitro on neuronal SH-SY5Y human cells “attacked” with a strong oxidizer (aqueous H_2_O_2_) [135];Naringenin (NRG) is a polyphenolic phytochemical belonging to the class of flavanones widely found in citrus fruits and other fruits such as cherries, tomatoes, bergamot and cocoa. NRG presents attractive pharmacological properties and potential therapeutic applications: it is antineoplastic, anti-oxidant, and anti-inflammatory. Different nanomedicine formulations of NRG are used to optimize the pharmacological properties of NRG and improve its delivery to the targeted organs/cell populations [136];Sulforaphane (SFN) is a nutrigenomic substance found in broccoli and other cruciferous vegetables that has anti-inflammatory and antioxidant effects. SFN can activate Nrf2 (transcription factor nuclear factor erythroid 2-related factor 2), induce antioxidant enzymes, and reduce proinflammatory cytokines. Nrf2 is an essential nuclear transcription factor, the “master regulator” that modulates detoxifying and antioxidant defense gene expression in the liver and the body’s antioxidant responses [137]. Sulforaphane-containing phytonutrients represent promising alternatives to traditional drugs as adjuvants in immune-associated NDs such as Alzheimer’s and Parkinson’s, promoting neuroprotection against COVID-19 [138];Anthocyanins represent promising natural products, adjuvants in NDs and PD, and adjuvants in a holistic therapeutic approach. Elderberry fruits have antiviral effects against SARS-CoV-2 and neuroprotective effects in PD. Blackcurrant anthocyanins increase the cellular glycine-proline concentration and have neuroprotective effects by improving IGF-1 function (insulin-like growth factor 1), which inhibits DA neurotoxicity and protects neurons [139];The biochemical, pharmacodynamics, and medicinal properties of cannabis plants are controversial, approved, or forbidden by different socio-economic, political, and legislative frames.


The principal chemicals are delta-9 tetrahydrocannabinol (THC), cannabidiol (CBD), cannabigerol (CBG), cannabichromene (CBC), cannabidivarin (CBDV), tetrahydrocannabivarin (THCV).

These compounds activate the endogenous cannabinoid receptors type 1 (CB1) and/or type 2 (CB2), the G protein-coupled receptor 55 (GPR55), peroxisome proliferator-activated receptor (PPAR), glycine receptors, serotonin receptors (5-HT), transient receptor potential channels (TRP), GPR and opioid receptors [123].

CB1 receptor activation is triggered by THC and is associated with unacceptable psychoactive responses (besides its neuromodulatory and analgesic effects).

The CB2 receptor is associated with anti-inflammatory and immunomodulatory effects. It regulates motor functions and dopamine activity in PD and has no psychoactive effects.

Cannabinoids are unconventional immunomodulatory agents. The cannabis constituents THC and CBD inhibit T-helper type 1 (Th1) cytokines and/or promote (in vitro and in vivo) the Th2 immune response [140].

Extracts with high content of CBD can downregulate ACE2 and transmembrane serine protease 2 (TMPRSS2), forbidding SARS-CoV-2 from entering the human cells through the upper respiratory tract. CBD might be a prophylaxis against COVID-19 but not administered by smoking [123,141];
C.Adjunctive medication:

N-acetylcysteine (NAC) as adjunctive therapy might be protective in neurodegenerative disorders. According to the antioxidant capacity of thiols, NAC might be useful in PD and prevent dopamine-induced cell death (both in vivo and in vitro).

Advanced age is associated with reduced glutathione levels, and NAC might be helpful in geriatric pathology [142]. NAC was able to modulate the immune response and possibly reduce COVID-19’s morbidity and mortality. Thiols can block the ACE2 receptor for the viral spike protein, inhibiting the penetration of SARS-CoV-2 into cells [143];
D.Immunomodulation:

The Bacille Calmette–Guérin vaccine (BCG) can nonspecifically modulate the immune system, inducing off-target immune effects in various pathologies. BCG rappel in adulthood might be an efficient alternative in reducing age-related events and the risk for NDs. It was associated with a 28% reduced risk of developing PD in vaccinated persons, respectively, and a 58% lower risk for Alzheimer’s disease [15].

The World Health Organization did not recommend a BCG vaccination to prevent COVID-19 [144];
E.Immunotherapeutic approach:

Novel therapies targeting fundamental pathogenic mechanisms involved in brain neuroinflammation and neurodegenerative disorders represent the field of research for many pharmaceutical companies worldwide.

(a) LRRK2 inhibitors

The leucine-rich repeat kinase 2 (LRRK2) gene and aSyn gene (SNCA) are the key pathogenic factors of PD. Complex relationships among LRRK2, aSyn and activated microglia are reported [37].

Discovering LRRK2 inhibitors, including DNL201 (Denali), represents challenges to PD clinical trials [108,145];

(b) Immunotherapeutic approaches targeting aSyn pathology:

Novel immunotherapeutic agents targeting extra-/or intracellular pathological aSyn might be efficient in limiting intracellular aggregation and cell-to-cell transmission in a prion-like manner.

Monoclonal antibodies (created by BioRender.com (accessed on 15 January 2022)) address cell surface receptors (i.e., LAG3), thus preventing the internalization of pathological aSyn. Lymphocyte activation gene-3 (LAG-3) represents an immune checkpoint protein-receptor on the T cells’ membrane, targeted by various drug development programs.

Monoclonal antibody-based therapies can block the extracellular aSyn, secreted by degenerated neurons in the blood and CSF.

Neuroinflammation is addressed with antibodies against pro-inflammatory cytokines (i.e., IL-1β and TNF-a) to prevent the activation of microglia and monocytes.

The immune complexes are internalized by microglia through the FCγRIII (vital in both acute and chronic inflammation), then degraded intracellularly.

Monoclonal antibodies enter neurons through the FCγRI and bind to the pathologic aSyn to stimulate its degradation.

Engineered nanobodies penetrate degenerated dopaminergic neurons, “polluted” with piles of pathological misfolded and/or mutated aSyn aggregates, and stimulate the degradation of Lewy bodies [91].

Some newly registered trials on immunotherapy agents targeting aSyn complex pathology are summarized below:−An oral, brain-penetrant inhibitor of aSyn aggregation (anle138b) was developed by the German biotech firm MODAG;−A monoclonal antibody targeting the pathological spread of aSyn aggregation was registered by AstraZeneca (MEDI1341/NCT04449484);−An aSyn-targeted vaccine (PD01A) was projected by AFFiRiS (no study being registered)−A phase 2 study (PASADENA, NCT03100149) project focusing on prasinezumab was launched by Roche and is in an expanded phase 2 program (NCT04777331);−aSyn antibodies BIIB054 (NCT03318523) created by Biogen, respectively venglustat (NCT02906020) produced by Sanofi started a phase 2 trial during 2021 [125].

Epigenetic related non-coding RNA (ncRNA) is a class of functional RNAs, which are fine-tuning regulators of gene expression at the transcriptional and post-transcriptional levels. They include miRNA, siRNA, piRNA, and lncRNA. Bringing miRNA-based therapeutics from bench to clinic might be helpful against the multiple neurodegenerative mechanisms in PD. The lethal-7 (let-7) gene, the first known microRNA, is a critical developmental regulator and might be a promising therapeutic tool targeting PD [115,146];
F.Nanotechnology and nanoparticles:

Nano-biomedical science and nanotechnology procedures might have therapeutic applications in the better delivery of cargo molecules to specific organs [147].

Nanoparticles (NPs) are used as nanocarriers for therapeutic agents and delivery systems. NPs include polymeric microparticles, micelles, liposomes, solid lipid nanoparticles (SLNs), nanostructured lipid carriers (NLCs), nanosuspensions, nanoemulsions, lipid-based, polymer-based, lipid–polymer hybrid-based, carbon-based, inorganic metal-based, surface-modified, and stimuli-sensitive nanomaterials [136,147].

Antiviral nanoformulation designs allowed overcoming the biological and biophysical barriers by improving the pharmacodynamics, pharmacokinetic properties, and biodistribution of the cargo content [147].

Microbiology and high technology are convergently progressing to NP-targeted antivirals used as vehicles and delivery systems of antiviral nanoformulations (such as long-acting antiviral NPs, cell- and tissue-targeted antiviral NPs, combinational antiviral NPs, and nucleic acid-based antiviral NPs) [147].

In the diagnosis, prevention, and therapeutics of viral infections, futuristic novel delivery systems include nano-traps, nanorobots, nanobubbles, nanofibers, and nanodiamonds [147].

The nanomaterials used for virus detection and tracking include magnetic and gold nanoparticles, ZnO/Pt-Pd, graphene, and quantum dots (QDs) [148,149].

Clinical trials are currently ongoing to assess the efficacy of inhalable anti-COVID-19 nanoformulations, such as remdesivir NPs and liposomal lactoferrin (Clinical Trial # NCT04480333 and # NCT04475120) [150].

Silver nanoparticles (AgNPs) were proposed as therapeutic alternatives against SARS-CoV-2. AgNPs bind to the spike glycoprotein through disulfide bonds, inhibit viral interactions with ACE2 receptors and hinder viral internalization into the host cell. AgNPs are captured into the cells through endocytosis and macropinocytosis, inhibiting the virus’s replication capacity.

Considering the risk of silver poisoning and argyrism, the US FDA does not recognize AgNPs as a safe and effective anti-COVID-19 agent. Inhalation was suggested as an alternative mehod, with lower adverse toxic effects [151].

Synthetic organoselenium compounds (ebselen, diphenyl diselenide) have specific inhibitory effects in some thiol-containing proteins and a non-specific modulatory activation of antioxidant genes.

Ebselen has a significant pharmacological potential to treat SARS-CoV-2 infection, the most potent inhibitor of SARS-CoV-2 replication both in vitro and in vivo. Ebselen inhibits thiol-containing viral proteases and decreases COVID-19 replication in vitro.

Selenium compounds have a small “window” of safe concentrations between the pharmacological effects and adverse toxicological reactions. The safe plasma levels are 80–120 mcg/L of selenium [111]. Its efficacy in animal models of PD was controversial. Organoselenium compounds (diphenyl diselenide and ebselen) can accentuate lipid peroxidation, induce high concentrations of ROS and thiol depletion, inhibit sulfhydryl enzymes, disrupt mitochondrial homeostasis, and trigger cellular toxicity and apoptosis [111].

## 5. Conclusions

The present article takes a holistic approach regarding the concepts of neurodegeneration and Parkinson’s disease in the COVID-19 pandemic era in an integrative frame: from the genomics perspective (and the inherited errors, with their molecular and cellular repercussions) to exposure to exo- and endogenous dysregulating factors of the individual’s bio-ecosystem. This complex “puzzle” is involved in the onset, progression, and long-persistent consequences of neurodegenerative diseases. The following information concludes our literature search:Angiotensin-converting enzyme 2 (ACE2) plays an essential role as the “Achilles’ heel” for the (neuro)-viral invasion and initiation point for the complex neurodegenerative mechanisms, facilitating viral penetration in the central and peripheral nerve structures and inducing neuronal and glial morpho-functional disorders;The acute cellular and humoral systemic inflammatory mechanisms exceed protection of the BBB, induce microglial activation, and trigger the pathogenic mechanisms leading to neurodegeneration mediated by neuroinflammation, secondary to COVID-19 infection and PD;Chronic neuroinflammation-mediated neurodegeneration is a significant factor for neuronal death, encountered in age-related brain pathology and long-term consequences of SARS-CoV-2 infection, including Parkinsonism-related symptoms;The intracellular and molecular pathophysiological interferences and neurodegenerative pathogenic consequences post-COVID-19 and PD refer to the disorders of the mitochondrial system and endocellular vesicular transport system and proteostasis as main targets and common pathways for self-sustained neuroinflammatory and neurodegenerative mechanisms;At the macroorganism level, the bio-psycho-neuro-immunological correlations between COVID-19 and PD are briefly presented, resulting in autonomic dysfunction as the main complaint in long COVID syndrome and PD dysfunctions;In this complex pathophysiological context, the integrative therapeutic management of the conventional arsenal and new unconventional approach to age-related neurodegeneration, PD and COVID-19 comprises engineered endogenous factors, exogenous phytonutrients, immunomodulatory, and nanoparticles as therapeutic support vehicles.

Time will confirm whether COVID-19 neuroinflammatory events might trigger long-term neurodegenerative effects and contribute to aggravation and/or explosion of new cases of Parkinson’s disease. The extent of this phenomenon remains obscure. Therefore, longitudinal prospective cohort studies are necessary.

## Figures and Tables

**Figure 1 biomedicines-10-01000-f001:**
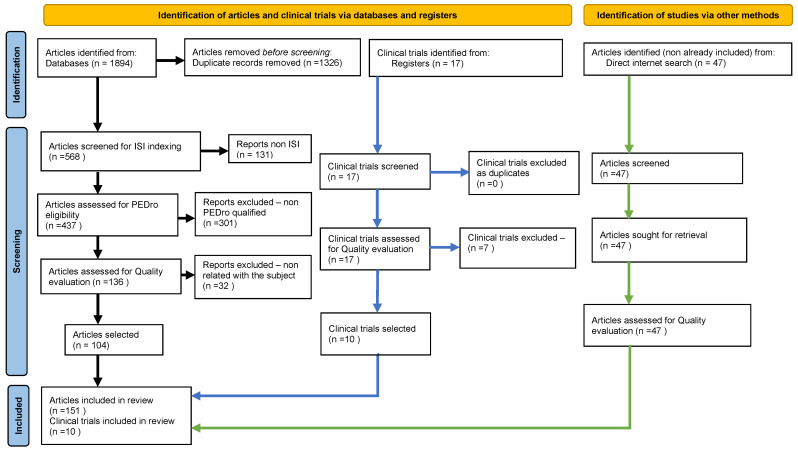
Our adapted PRISMA-type flow diagram.

**Figure 2 biomedicines-10-01000-f002:**
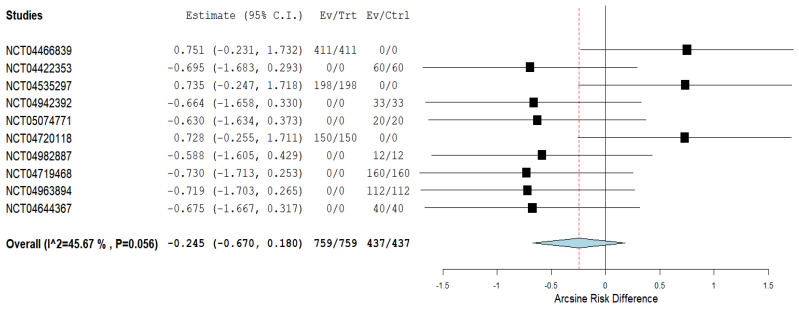
Meta-analysis of the data obtained from the identified clinical studies. Estimate lower-bound upper-bound std. error *p*-value −0.245 −0.670 0.180 0.217 0.258. Heterogeneity. tau^2 Q (df = 9) Het. *p*-value I^2. 0.215 16.567 0.056 45.674.

**Table 1 biomedicines-10-01000-t001:** The clinical trials that satisfied all the previous filtering criteria/PRISMA steps selected for qualitative synthesis were included in our meta-analysis.

NCT Number	Title	Interventions	Study Type:	Population
NCT04466839	Evaluation of the Containment Impact Linked to the COVID-19 Pandemic in a Population of Parkinson’s	• Other: questionnaire and interview	Observational	411
NCT04422353	Video Dance Class and Unsupervised Physical Activity During COVID-19 Pandemic in People with Parkinson’s Disease	• Other: video dance classes • Other: unsupervised physical activities	Interventional	60
NCT04535297	Consequences of the COVID-19 Lockdown on Health and Well-being of Patients with Parkinson‘s Disease and Post-stroke	• Other: exposure	Observational	198
NCT04942392	Digital Dance for People with Parkinson’s Disease During the COVID-19 Pandemic	• Device: digital dance for PD	Interventional	33
NCT05074771	At Home REhabilitation and Monitoring of People in poST-COVID Condition Through ARc-inTellicare Platform (RESTART/RICOMINCIARE)	• Device: ARC intellicare	Interventional	20
NCT04720118	Parkinson’s Disease and Experiences Throughout the COVID-19 Pandemic	• Other: no intervention	Observational	150
NCT04982887	Tele-Rehabilitation in Parkinson’s Disease	• Other: exercise	Interventional	12
NCT04719468	PD-Ballet: Effectiveness and Implementation in Parkinson’s Disease	• Other: dance with ballet elements	Interventional	160
NCT04963894	Effects of Home Rehabilitation of Balance Based on Functional Exercises in People with Parkinson’s Disease	• Other: home functional balance physiotherapy • Other: conventional physiotherapy	Interventional	112
NCT04644367	Effects of a Biomechanical-based Tai Chi Program on Gait and Posture in People with Parkinson’s Disease	• Other: Tai Chi intervention • Other: regular physical activity (control) group	Interventional	40

## Data Availability

Not applicable. This systematic review was submitted on the PROSPERO—International prospective register of systematic reviews—Online platform, No. 312183.

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
