# Peer review of "Parkinson’s Disease and SARS-CoV-2 Infection: Particularities of Molecular and Cellular Mechanisms Regarding Pathogenesis and Treatment"

_biomedicines, 2022, doi:10.3390/biomedicines10051000_

Round 1

Reviewer 1 Report

Thanks for recommending me as a reviewer. This systematic review is focused on a recent question aroused during the pandemic successive waves: are there post SARS-CoV-2 immune-mediated reactions responsible for promoting neurodegeneration, or does the host`s dysregulated immune counteroffensive contribute to the pathogenesis of neurodegenerative diseases, respectively emerging Parkinson's disease, in a complex interrelation between genetic and epigenetic risk factors. If authors complete minor revisions, the quality of the study will be further improved.

  1. A paragraph cannot consist of a single sentence. The third and fourth paragraphs of the introduction should be combined with the other paragraphs.

2. Are hyperlinks required in Table 3?

3. Did you not include quality assessment in this systematic review study? If included, authors should add results for quality assessment in Table 3.

Author Response

Dear reviewer, the authors warmly thank you for your highly professional and detailed analysis of our article, which helped us improve our manuscript. Regarding your comments and suggestions, please find herein below our answers, point-by-point:

1. A paragraph cannot consist of a single sentence. The third and fourth paragraphs of the introduction should be combined with the other paragraphs.

You are perfectly right. This issue was properly resolved.

2. Are hyperlinks required in Table 3?

Because they are links to clinical trials and not published papers – and especially as they are not many (only ten) we consider it appropriate to keep the respective links in table 3 and help readers to find the reports.

3. Did you not include quality assessment in this systematic review study? If included, authors should add results for quality assessment in Table 3.

Actually, for the largest majority of bibliographic resources, i.e published articles we did the qualitative analysis: PEDro inspired method and scoring (kindly see Supplementary material). As regards only the clinical trials we had included in our literature review, as above mentioned, we found only ten of them to be contributive – others being excluded as their content was not limited to this pathology only and we wanted to avoid tangentially interferences to our search.

Reviewer 2 Report

The review of Anghelescu et al is an effort to combine experimental findings to address whether COVID19 infammation could impact neurodegeneration in Parkinson's disease (PD). The procedure followed for the meta-analysis is clearly described. The study resulted in highlighting 11 successive steps that cover some of the most important aspects in the PD field with a special emphasis, as expected, in neuroinflammation. My only concern is that the review is rather long and could be difficult to follow as a whole. One point to suggest is to enrich the conclusions section by combining the information presented. In other words, to state what are the most significant correlations found after COVID19 pandemic and how these can affect/change PD research.

Author Response

Dear reviewer, the authors warmly thank you for your professional and detailed analysis of our manuscript and for your constructive suggestion for improving it. This systematic review is focused on a recent question aroused during the pandemic successive waves: are there post SARS-CoV-2 immune-mediated reactions responsible for promoting neurodegeneration, or does the host`s dysregulated immune counteroffensive contribute to the pathogenesis of neurodegenerative diseases, respectively emerging Parkinson's disease, in a complex interrelation between genetic and epigenetic risk factors. For responding to the assumed task, we searched in various databases and obtained initially 1894 articles. After fulfilling the five steps in the selection methodology, 104 papers were selected for synthetic and systematic review. The subject is an integrated one and we tried to reduce the information in the manuscript as much as possible. Still, considering your suggestion, we enriched the Conclusion section with correlations discussed in the previous section.

Round 2

Reviewer 1 Report

Thanks for recommending me as a reviewer. In this systematic review, authors were focused on a recent question aroused during the pandemic successive waves: are there post SARS-CoV-2 immune-mediated reactions responsible for promoting neurodegeneration, or does the host`s dysregulated immune counteroffensive contribute to the pathogenesis of neurodegenerative diseases, respectively emerging Parkinson's disease, in a complex interrelation between genetic and epigenetic risk factors. If authors complete minor revisions, the quality of the study will be further improved.

  1. Authors should change the style of the manuscript to the style of the journal.

2. Authors must submit final files with English proofreading.

3. Figure 1. Right Column, If there are no excluded papers, it is recommended to delete them from the figure.

4. Quality evaluation was omitted in this study. Where possible, authors should add a quality assessment.